# Effects of *Bacillus licheniformis* on Growth Performance, Diarrhea Incidence, Antioxidant Capacity, Immune Function, and Fecal Microflora in Weaned Piglets

**DOI:** 10.3390/ani12131609

**Published:** 2022-06-22

**Authors:** Xiaorong Yu, Zhenchuan Cui, Songke Qin, Ruiqiang Zhang, Yanping Wu, Jinsong Liu, Caimei Yang

**Affiliations:** 1College of Animal Science and Technology, College of Animal Medicine, Zhejiang Agriculture & Forestry University, Hangzhou 311300, China; yuxiaorong@stu.zafu.edu.cn (X.Y.); czc@stu.zafu.edu.cn (Z.C.); 2020608022027@stu.zafu.edu.cn (S.Q.); zrq1034@zafu.edu.cn (R.Z.); 2Key Laboratory of Molecular Animal Nutrition (Zhejiang University), Ministry of Education, Hangzhou 311300, China; ypwu@zafu.edu.cn; 3Institute of Animal Health Products, Zhejiang Vegamax Biotechnology Co., Ltd., Huzhou 313306, China; 13906510961@163.com

**Keywords:** *Bacillus licheniformis*, growth performance, diarrhea incidence, immune function, fecal microflora, piglets

## Abstract

**Simple Summary:**

*Bacillus licheniformis* has been shown to be safe as a green additive in food and feed. This experiment was conducted to investigate the value of *Bacillus licheniformis* in the diet of piglets. Our results suggested that dietary *Bacillus licheniformis* supplementation plays an important role in improving the average daily gain, alleviating diarrhea, improving antioxidant capacity, promoting immune function, and regulating the intestinal microflora of weaned piglets.

**Abstract:**

*Bacillus licheniformis* (*B. licheniformis*) is a safe probiotic that can promote animal growth and inhibit pathogenic bacteria. This study aimed to assess the effects of *B. licheniformis*, one green feed additive, on growth performance, diarrhea incidence, immune function, fecal volatile fatty acids, and microflora structure in weaned piglets. Weaned piglets (*n* = 180) were randomly divided into three treatment groups and fed a basal diet and a basal diet supplemented with 500 mg *B. licheniformis* per kg and 1000 mg *B. licheniformis* per kg, respectively. The dietary 500 mg/kg *B. licheniformis* inclusion improved the average daily gain, reduced diarrhea incidence, and strengthened antioxidant capacity. Piglets supplemented with *B. licheniformis* presented increased serum immunoglobulins (IgA, IgM) compared to the CON group. Meanwhile, the expression of anti-inflammation factors was increased, and the levels of pro-inflammation factors were reduced after *B. licheniformis* administration. Moreover, the levels of volatile fatty acids, including acetic acid, propionic acid, butyric acid, isobutyric acid, and isovaleric acid, in the BL500 and BL1000 groups were increased compared with the CON group, and the concentration of valeric acid was higher in the BL500 group. Furthermore, piglets in the 500 mg/kg *B. licheniformis* addition group significantly altered fecal microbiota by increasing *Clostridium_sensu_stricto_1* and *Oscillospira*. In conclusion, dietary *B. licheniformis* relieved diarrhea, enhanced antioxidant capacity, immunity function, and fecal microflora structure in weaned pigs.

## 1. Introduction

Early weaning is a stress, and pigs commonly experience diarrhea and a reduced growth rate during postweaning [1]. Antibiotics are usually used to enhance the immune response and command the incidence of many diseases in weaned piglets [2]. However, in recent years, antibiotic resistance has become an important global issue and has received widespread concern in the livestock industry [3]. Antibiotic resistance has an undesirable effect on people other than the immediate consumer of the antibiotic [4]. Several research studies on promising alternatives to antibiotics are currently being developed in livestock, such as probiotics [5], acidifiers [6], plant extracts [7], and so on.

Several studies have demonstrated that probiotics are beneficial for improving growth performance and immune function in animals [8]. Among the prebiotics and probiotics commonly adopted, *B. licheniformis* is a spore-forming Gram-positive bacterium that has been used in animal feed for a long time [9]. *B. licheniformis* can produce bacitracin against pathogenic microorganisms [10]. In vivo experiments, *B. licheniformis* has the function of promoting growth and acting as a competitive repellent, stimulating appetite, promoting digestion, and increasing nutrient retention and absorption [11]. Research demonstrated that *B. licheniformis* repaired the imbalance of intestinal flora in deoxynivalenol-challenged mice [12]. *B. licheniformis* strain CK1 had probiotic characters, including acid resistance, adhesion ability, and resistance to pathogenicity [13]. Supplementation in the diet with *B. licheniformis* could increase digestibility and fecal *Lactobacillus* counts and reduce fecal toxic gas emissions in growing-finishing pigs [14]. Lin et al. [15] added 0.5 g/kg of a *B. licheniformis*-fermented feed additive, containing 5 × 10^11^ CFU/g *B. licheniformis*, to the basal diet in a weaned piglet experiment. Xu et al. [16] explored the role of *B. licheniformis* in broiler chickens, where the *B. licheniformis* group was fed a basal diet supplemented with 1.5 × 10^9^ CFU/kg *B. licheniformis*. Based on previous studies, we set the dose of *B. licheniformis* for weaned piglets.

However, few studies have evaluated whether *B. licheniformis* was a promoter for weaned pig growth and health. The current study aimed to prove the hypothesis that the application of *B. licheniformis* in piglets’ diet could alleviate the diarrhea incidence, promote growth performance, antioxidant capacity, immune function, and alter the fecal microbial composition in weaned piglets. The combined analysis of these results may help to find specific microorganisms, as well as provide insights into the potential microbial mechanisms of piglets supplemented with *B. licheniformis*.

## 2. Materials and Methods

### 2.1. Animals and Feedings

The current study was carried out according to the regulations by the Animal Care and Use Committee of the Zhejiang A&F University (No. ZAFUAC2021026). A total of 180 four-week-old weaned Duroc × Landrace × Yorkshire piglets were raised in Zhengxing Animal Husbandry Co., Ltd. (Hangzhou, Zhejiang, China), and the piglets were randomly divided into three treatment groups (6 replicates, 10 piglets per pen) as follows: CON (basal diet), BL500 (basal diet supplemented with 500 mg *B. licheniformis* per kg of basal diet), and BL1000 (basal diet supplemented with 1000 mg *B. licheniformis* per kg of basal diet). The corresponding amounts of *B. licheniformis* were evenly mixed into the basic diet for feeding. The experiment lasted for 28 days. The *Bacillus licheniformis* product was used at a concentration of 1 × 10^9^ CFU per gram of *B. licheniformis*. The strain has been saved in the China General Microbiological Culture Collection Center (CGMCC) and its collection number is HJ0135. The basal diet for piglets was designed to conform to the nutritional necessities of NRC (2002) without antibiotics (Table 1). Piglets were fed at a suitable temperature and water was provided ad libitum. The piglets were vaccinated according to the routine methods of the farm.

### 2.2. Sample Collection

On day 28, 18 piglets were chosen to collect blood samples from the front cavity vein, with body weight close to the average at Zhengxing Animal Husbandry Co., Ltd. (Hangzhou, Zhejiang, China). Blood samples were collected and centrifuged (3000× *g* for 15 min at 4 °C for serum separation. Furthermore, feces were also collected in a sterile tube to determine the composition of fecal microorganisms. Serum and fecal samples were stored at −80 °C for further analysis.

### 2.3. Growth Performance and Diarrhea Rate

Piglets from different groups were weighed individually at the start and the end of the experiment to monitor body weight and calculate the average daily gain (ADG). Feed intake was recorded each day in each pen. The average daily feed intake (ADFI) and feed to gain ratio (F:G) were computed using the recorded data. The number of pigs with diarrhea was observed and recorded daily to evaluate the incidence of diarrhea. The diarrhea rate (%) = total number of diarrhea piglets/(total number of piglets × test days) × 100.

### 2.4. Serum Antioxidant Indexes

The serum antioxidant indexes, including glutathione peroxidase (GSH-Px), total antioxidant capacity (T-AOC), superoxide dismutase (SOD), and malondialdehyde (MDA), were determined in accordance with the manufacturer’s instructions using relevant diagnostic kits purchased from the Angle Gene Bioengineering Co., Ltd. (Nanjing, Jiangsu, China). The Lot number of SOD was 202204011C, the Lot number of GSH-Px was 202204033C, the Lot number of MDA was 202204052D, and the Lot number of T-AOC was 202204024B.

### 2.5. Immune Indexes

Concentrations of immunoglobulin (IgG, IgA, IgM) and inflammatory cytokines (interleukin (IL)-6, IL-1β, IL-10, and tumor necrosis factor (TNF)-α) were determined using enzyme-linked immunosorbent assay (ELISA) kits. The kits above were bought from Angle Gene Biotechnology Co. Ltd. (Nanjing, Jiangsu, China). The specific test steps followed the manufacturer’s instructions. The Lot number of IgG was 202111062B, the Lot number of IgG was 202111051C, the Lot number of IgM was 202111032C, the Lot number of IL-1β was 202111063A, the Lot number of IL-10 was 202111041B, the Lot number of TNF-α was 202111063D, and the Lot number of IL-6 was 202111044A.

### 2.6. Fecal Volatile Fatty Acids (VFAs)

Fecal VFAs were assayed by a Gas Chromatography System (7890B, Agilent Technologies, Wilmington, DE, USA) according to previous studies [17]. In brief, the samples (0.5 g) were mixed with 1 mL of pre-cooled deionized water and centrifuged (12,000× *g* for 10 min at 4 °C) to gain the supernatant. Then, the supernatant was mixed with 25% metaphosphoric acid (*w*/*v*) at a ratio of 5:1, and then centrifuged (12,000× *g* for 10 min at 4 °C) to obtain the supernatant. After centrifugation, approximately 500 μL of the supernatant was filtered into sample bottles for further gas chromatography analysis 

### 2.7. Fecal Flora Structure

The fecal samples from different piglets were used for the microbiome analysis. The genomic DNA of fecal microbial was extracted using the DNeasy Power Soil Kit (Qiagen, Hilden, Germany) following the manufacturer’s instructions. The specific primer sequences 515F (5′-GTGCCAGCMGCCGCGGTAA-3′) and 806R (5′-GGACTACHVGGGTWTCTAAT-3′) were used for the analysis of V3-V4 region of 16S rRNA gene. The sequencing was conducted on the Illumina Miseq platform (Illumina, San Diego, CA, USA), and the data were analyzed on the Majorbio Cloud Platform, an online platform provided by Majorbio Bio-Pharm Technology Co., Ltd. (Shanghai, China). The amplicon sequence variant (ASV) is a set composed of concerned species. These concerned species can be comprised of unique or common species information in different groups obtained through a Venn diagram. ASV can be used for composition analysis and correlation analysis of specific species. Alpha diversity analyses were used to describe the degree of fecal microflora diversity between each group. 

### 2.8. Statistical Analysis

All data were checked by the equality of variance with normality distribution, then analyzed with SPSS 25.0 software (SPSS Inc., New York, NY, USA) using a one-way ANOVA to analyze whether the species diversity was significant between three independent experiments with at least six replicates per independent experiments. The histograms were created using GraphPad Prism 8 software (GraphPad Prism Inc., San Diego, CA, USA). 0.05 < *p* < 0.1 was considered a trend and *p* < 0.05 was considered a significant difference.

## 3. Results

### 3.1. Growth Performance and Diarrhea Rate

The effects of *B. licheniformis* on growth performance in weaned piglets are indicated in Table 2. Compared with the control group, 500 mg/kg *B. licheniformis* supplementation significantly enhanced the ADG in piglets throughout the entire experimental period (*p* < 0.05). Dietary supplementation with 500 and 1000 mg/kg of *B. licheniformis* could significantly reduce the diarrhea rate of piglets (*p* < 0.05).

### 3.2. Serum Antioxidant Indexes

The effects of *B. licheniformis* on the serum antioxidant parameters in weaned piglets are shown in Figure 1. Compared to the CON group, dietary 500 and 1000 mg/kg *B. licheniformis* significantly increased the serum GSH-Px activity and decreased the MDA level of serum in weaned piglets (*p* < 0.05). Moreover, the supplementation of 500 mg/kg of *B. licheniformis* significantly improved the serum levels of T-AOC and SOD (*p* < 0.05). In addition, dietary 500 mg/kg *B. licheniformis* markedly improved the serum GSH-Px activity and reduced the MDA level of serum in weaned piglets compared to the BL1000 group (*p* < 0.05).

### 3.3. Serum Immune Cytokine

As demonstrated in Figure 2, compared with the control piglets, 500 mg/kg *B. licheniformis*-treated piglets exhibited significantly higher serum IgA and IgM concentrations and lower serum IL-1β and IL-6 concentrations (*p* < 0.05). The level of serum anti-inflammatory factor IL-10 in the BL500 and BL1000 groups was higher than that in the CON groups (*p* < 0.05).

### 3.4. Concentrations of VFAs

The results of the VFA analysis are shown in Figure 3. The concentrations of acetic acid, propionic acid, isobutyric acid, butyric acid, isovaleric acid, and valeric acid dietary were significantly higher in the BL500 group compared with the control group (*p* < 0.05). Moreover, the levels of acetic acid, propionic acid, isobutyric acid, butyric acid, and isovaleric acid markedly increased in the BL1000 group compared to the CON group (*p* < 0.05). Compared with the BL1000 group, the contents of acetic acid, isobutyric acid, and isovaleric acid were significantly higher in the BL500 group (*p* < 0.05).

### 3.5. Microflora Structure in the Colonic Contents

The microbiota in the colonic content is summarized in Figure 4. A total of 725 ASVs were shared among the three treatment groups (Figure 4A). The control group piglets had 463 unique ASVs, BL500 group piglets had 388 ASVs, and the BL1000 group had 341 unique ASVs (Figure 4A). The Chao index represents the community richness of fecal samples, while the Simpson index reflects the community diversity of fecal samples. The addition of 500 mg/kg *B. licheniformis* gave lower Chao index and higher Simpson index readings for fecal microbiota in piglets and 1000 mg/kg *B. licheniformis* addition showed a lower Chao index (*p* < 0.05, Figure 4B,C). 

At the phylum level, Firmicutes, Bacteroidota, and Spirochaetota were the three major phyla (Figure 4D). At the family level, Prevotellaceae, Clostridiaceae, and Lachnospiraceae were the dominant flora (Figure 4E). The addition of 500 mg/kg of *B. licheniformis* significantly increased the content of Clostridiaceae (*p* < 0.05, Figure 4F). At the genus level, *Prevotella Norank-F-Muribaculaceae*, *Clostridium sensu stricto 1*, *Prevotellacea-NK3B31-group*, *Lactobacillus*, *Terrisporobacter*, *unclassified_f__Lachnospiraceae*, *Rikenellaceae_RC9_gut_group*, *UCG-005*, and *Phascolarctobacterium* were the top 10 genera in abundance values (Figure 4G). Compared with the CON group, adding 500 mg/kg of *B. licheniformis* could dramatically increase the contents of *Clostridium_sensu_stricto_1* and *Oscillospira* (*p* < 0.05, Figure 4H,I). 

### 3.6. Spearman’s Correlation Analysis

The results of Spearman’s correlation analysis between fecal microbiota and serum immune cytokines are shown in Figure 5. The genera *norank_f__Prevotellaceae* and *Coprococcus* had a significant positive correlation with the pro-inflammatory factor IL-6, while *Selenomonas* and *Megasphaera* had a significant negative correlation with the anti-inflammatory factor IL-10 (*p* < 0.05). What is more, heatmaps of Spearman’s correlation analysis between fecal microbiota and volatile fatty acids are shown in Figure 6. The genera *Oscillospira* had an observable positive association with acetic acid, propionic acid, isobutyric acid, and isovaleric acid; meanwhile, *Clostridium_sensu_stricto_1* was positively correlated with valeric acid (*p* < 0.05). However, *Megasphaera* and *Selenomonas* had significant negative relations with acetic acid, propionic acid, isobutyric acid, isovaleric acid, and valeric acid (*p* < 0.05).

## 4. Discussion

Pathogen invasion and environmental stress affected the incidence of diarrhea by changing the composition of intestinal flora [18]. Weaning stress, one of the environmental stresses, can easily cause diarrhea in piglets, which contributes to intestinal inflammation. *Bacillus* strains could improve infectious enteritis or diarrhea in weaned piglets [19]. Similarly, the present study results indicated that *B. licheniformis* significantly reduced the incidence of diarrhea. According to Lin et al. [15], a *B. licheniformis*-fermented feed additive not only reduced the incidence of diarrhea, but also tended to increase body weight and the average daily gain of weaned piglets. In this study, supplementation of the diet with 500 mg/kg of *B. licheniformis* markedly enhanced the ADG compared with the control group, while a diet with *B. licheniformis* could increase ADFI and reduce F:G but the effects were not significant. What is more, there was no significant difference in growth performance between the BL500 group and the BL1000 group. The improved growth performance of piglets fed *B**. licheniformis* may be related to beneficial metabolites produced by *B. licheniformis*, such as extracellular digestive enzymes and lysozymes [20]. It is also possible that feeding *B. licheniformis* could improve the immunity of piglets [21]. *B. licheniformis* reduced the pathogens in the gut through competitive rejection, while increasing the uptake of nutrients in the intestine, as it was beneficial to improving the growth performance of animals [22]. The ADG of piglets in the BL500 group was significantly higher than the CON group, indicating that a lower dose of *B. licheniformis* was better for modulating the gut health of piglets, but the mechanism needs to be further explored. Piglets fed a diet with *B. licheniformis* showed reduced damage to the villus height and increased the villus height: crypt depth in the duodenum, jejunum, and ileum [23]. Liu et al. [8] found that *B. licheniformis* had positive effects on controlling *Salmonella* Infantis infection via alleviating inflammation and maintaining the intestinal mucosal barrier integrity in pigs.

Researchers hypothesize that oxidative stress affects growth performance in animals [24]. The antioxidant capacity of the host was evaluated by measuring catalase, SOD, GSH-Px, and other related enzymes [25]. Oxidative stress can produce large amounts of MDA, which ultimately leads to tissue damage and the development of diseases. Xu et al. [26] showed that weaning induced oxidative stress characterized by increasing the contents of MDA and free radicals and impairing the antioxidant defense system of piglets. T-AOC refers to the total antioxidant level composed of various antioxidant substances and antioxidant enzymes [27]. GSH-Px is a significant part of the enzyme antioxidant defense system [28]. Peng et al. [29] found the supplementation of *B. licheniformis* and *Saccharomyces cerevisiae* (*S. cerevisiae*) increased the activities of SOD and GSH-Px. In this study, compared with the CON group, *B. licheniformis* improved the antioxidant capacities in the serum of piglets by increasing T-AOC, GSH-Px, and SOD activities and by decreasing MDA. Similarly, *B. licheniformis* enhanced muscle antioxidant capacity and improved meat quality in late-finishing pigs [30]. 

Serum immunoglobulins and inflammatory cytokines are typical parameters for estimating the immune status of livestock and poultry [31]. Excess pro-inflammatory cytokines may reduce feed intake and increase energy expenditure, thus reducing the performance of livestock [32]. In the present case, adding 500 mg/kg of *B. licheniformis* could elevate IgA, IgM, and IL-10 levels and reduce the contents of IL-6 and IL-1β compared with control piglets. The results indicated that the lower dose of *B. licheniformis* could be more beneficial to enhance immune function, possibly because it could better regulate the balance of intestinal flora in piglets, while the exact mechanism needs to be further studied. Kwak et al. [33] found the levels of IL-1β and IL-12 in the jejunum of growing-finishing pigs were down-regulated by multispecies probiotic formulation supplementation. Spearman’s correlation analysis revealed the genera *norank_f__Prevotellaceae* and *Coprococcus* had a significant positive association with IL-6, while *Selenomonas* and *Megasphaera* had a significant negative relationship with the anti-inflammatory factor IL-10. However, Huang et al. [34] showed *Coprococcu**s* was negatively correlated with IL-17A, suggesting that it may be involved in pemphigus vulgaris by regulating T cell differentiation and the related cytokines. Indeed, previous data showed that the *Megasphaera* sp. *XA511* promoted the secretion of pro-inflammatory cytokines of peripheral blood mononuclear cells, suggesting that the *Megasphaera* might enhance the immune response [35]. The active triple peptides reduced the expression of inflammatory cytokines, thereby reducing the abundance of *Epsilonproteobacteria*, *Erysipelotrichales*, *Prevotellaceae*, *Flavobacteriaceae,* and other pathogenic bacteria [36].

The intestinal VFAs play an active role in nutrient metabolism and immune function, while also regulating the intestinal microbiota at the same time [37], obtaining the fermentation of polysaccharides by intestinal anaerobic bacteria [38]. VFAs also create an acidic environment in the gut to inhibit some harmful bacteria [39]. Xu et al. [16] indicated that the levels of VFAs, including butyric acid, isobutyric acid, valeric acid, and isovaleric acid, in fecal content were improved by adding *B. licheniformis*. Adding probiotics to the diet resulted in a higher proportion of VFAs, mainly acetic acid, which may be beneficial to the intestinal environment and may reduce post-weaning diarrhea in piglets [40]. These findings were similar to this study, whereby the *B. licheniformis* supplementation induced an incremental increase in fecal VFAs. This result is consistent with the previous research where higher VFAs concentrations were reported because of *B. licheniformis* supplementation [41]. Spearman’s correlation analysis showed that *Oscillospira* was positively associated with acetic acid, propionic acid, isobutyric acid, and isovaleric acid, and *Clostridium sensu stricto 1* was positively related to valeric acid. Yang et al. [42] found that *Oscillospira* could produce various VFAs, mainly butyric acid. Moreover, *Clostridium_sensu_stricto_1* and *Clostridium_sensu_stricto_4* were responsible for the change in butyric and propionic acid contents in the colon of anemic animals [43]. 

Various research had interpreted a close link between fecal microbes, VFAs, and immune function [44,45]. Richards et al. [46] indicated that animal intestinal flora had nutritional and protective functions, activated the immune system, produced beneficial products in intestinal fermentation, and prevented pathogen colonization. Earlier studies had shown that Firmicutes, Bacteroidetes, Proteobacteria, and Actinobacteria were the most dominant phyla in piglets [47]. In this case, Firmicutes and Bacteroidota were the advantage of bacterial phyla. At the genus level, the current results revealed a high abundance of *Clostridium_sensu_stricto_1* in the BL500 group compared with the control group. *Clostridium* sensu stricto has been associated with necrotizing enterocolitis, growth performance, and inflammation in pig models [48]. *Clostridium_sensu_stricto_1* and *Clostridium_sensu_stricto_2* were two major groups in the *Clostridium* genus. Compared with healthy piglets, the content of *Clostridium_sensu_stricto_2* was higher in piglets with diarrhea, whereas *Clostridium_sensu_stricto_1* exhibited contrary results, indicating *B. licheniformis* could reduce the diarrhea incidence by increasing the level of *Clostridium_sensu_stricto_1* [49]. In addition, *Clostridium_sensu_stricto_1* was reported to produce VFAs using mucus-derived sugars as an energy source and promote intestinal mucus barrier adhesion to pathogens [50]. Moreover, Fu et al. [51] found the abundance of *Clostridium_sensu_stricto_1* significantly increased in the resveratrol group. Through correlation analysis, Zhou et al. [52] showed the elevated relative abundances of *Clostridium_sensu_stricto_1* in cecum contents were negatively correlated with the level of inflammatory factors IL-12, IL-6, and IFN-γ. 

## 5. Conclusions

In summary, the supplementation of 500 mg/kg *B. licheniformis* in the diet of weaning piglets could increase the average daily gain, relieve diarrhea, enhance antioxidant capacity, improve immune function, and regulate the intestinal flora in weaned piglets. *B. licheniformis* may increase the genera *Oscillospira* and *Clostridium_sensu_stricto_1* to promote growth performance. These findings provide novel insight into *B. licheniformis* enhancing beneficial effects in weaned piglets.

## Figures and Tables

**Figure 1 animals-12-01609-f001:**
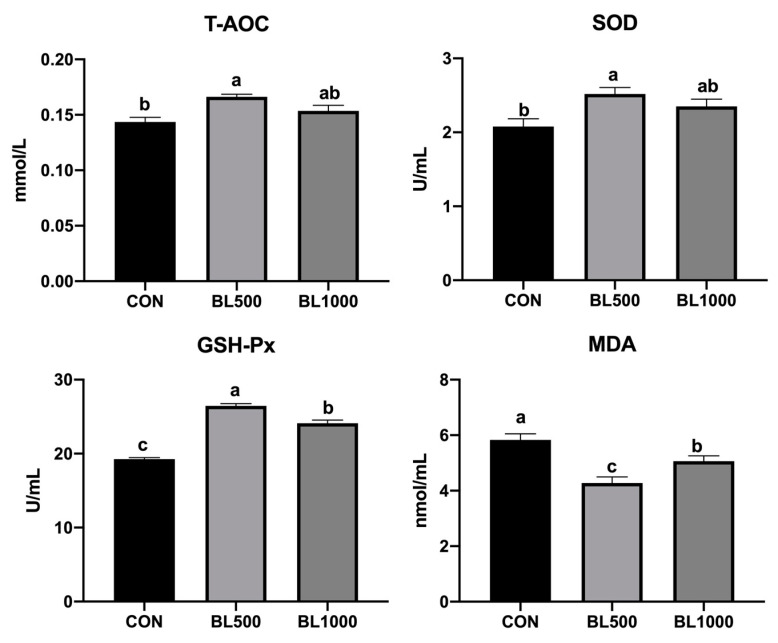
Effects of *B. licheniformis* on serum antioxidant parameters of weaned piglets. CON represents the control piglets; BL500 represents the piglets supplemented with 500 mg/kg *B. licheniformis*; BL1000 represents the piglets supplemented with 1000 mg/kg *B. licheniformis*. ^a,b,c^ Means with different superscripts in the same row differ significantly (*p* < 0.05). Note: T-AOC, total antioxidant capacity; SOD, superoxide dis-mutase; GSH-Px, glutathione peroxidase; MDA, malondialdehyde. *n* = 6.

**Figure 2 animals-12-01609-f002:**
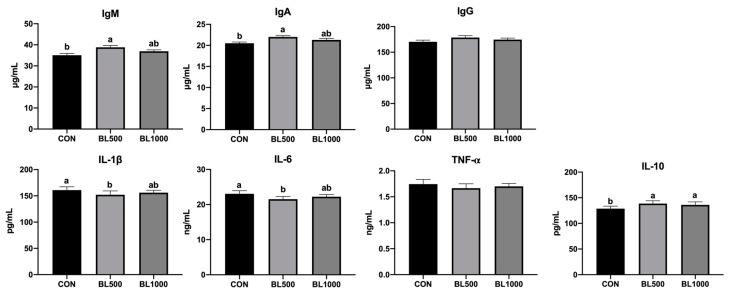
Effects of *B. licheniformis* on serum immune cytokine of weaned piglets. CON represents the control piglets; BL500 represents the piglets supplemented with 500 mg/kg *B. licheniformis*; BL1000 represents the piglets supplemented with 1000 mg/kg *B. licheniformis*. ^a,b^ Means with different superscripts in the same row differ significantly (*p* < 0.05). Note: IgG, immunoglobulin G. IgA, immunoglobulin A. IgM, immunoglobulin M. IL-6, interleukin-6. IL-1β, interleukin-1β. IL-10, interleukin-10. TNF-α, tumor necrosis factor-α. *n* = 6.

**Figure 3 animals-12-01609-f003:**
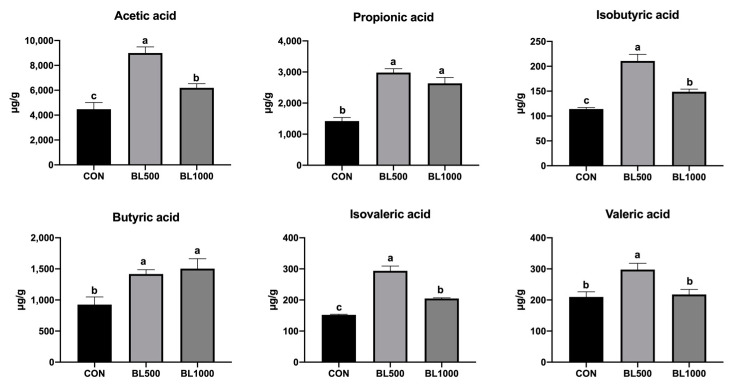
Effects of *B. licheniformis* on the colonic concentrations of VFAs in weaned piglets. CON represents the control piglets; BL500 represents the piglets supplemented with 500 mg/kg *B. licheniformis*; BL1000 represents the piglets supplemented with 1000 mg/kg *B. licheniformis*. ^a,b,c^ Means with different superscripts in the same row differ significantly (*p* < 0.05). *n* = 6.

**Figure 4 animals-12-01609-f004:**
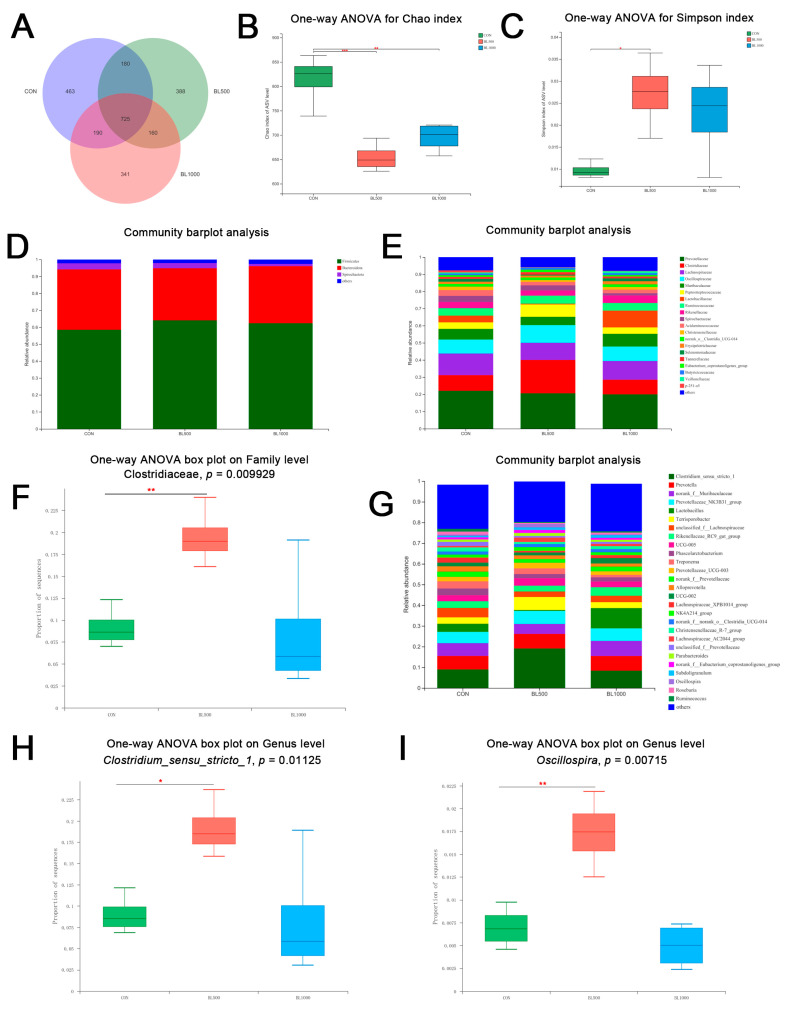
Summary of the microbial community composition in the fecal contents of weaned piglets. (**A**) Venn diagram summarizing the numbers of common and unique ASVs in the microflora community in the fecal in piglets. (**B**) Chao index. (**C**) Simpson index. (**D**) The microbiota composition on phylum level. (**E**,**F**) The microbiota composition on family level. (**G**–**I**) The microbiota composition on genus level. * *p* < 0.05, ** *p* < 0.01, *** *p* < 0.001. *n* = 6.

**Figure 5 animals-12-01609-f005:**
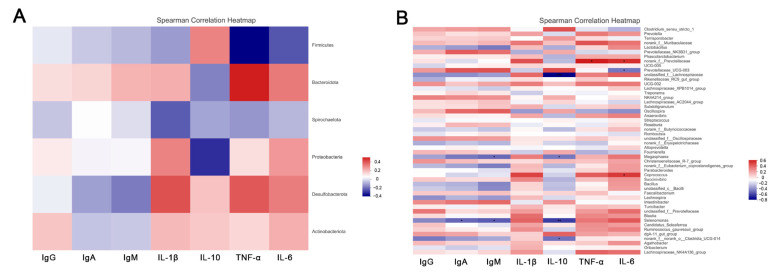
Heatmaps of Spearman’s correlation analysis between fecal microbiota and serum immune cytokines ((**A**): Phylum; (**B**): Genus. * *p* < 0.05, ** *p* < 0.01, *** *p* < 0.001).

**Figure 6 animals-12-01609-f006:**
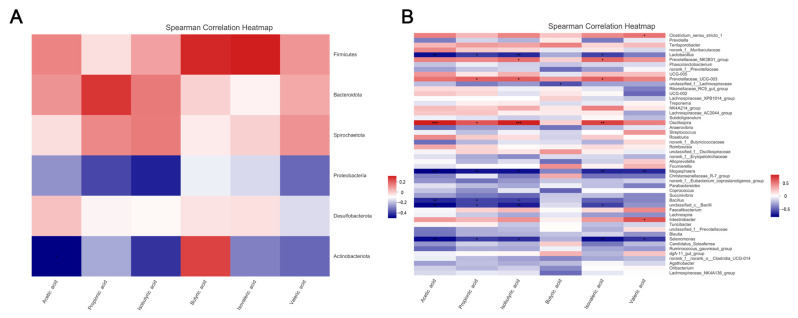
Heatmaps of Spearman’s correlation analysis between fecal microbiota and volatile fatty acids ((**A**): Phylum; (**B**): Genus. * *p* < 0.05, ** *p* < 0.01, *** *p* < 0.001).

**Table 1 animals-12-01609-t001:** Composition and nutrient levels of the basal diet (air-dry basis).

Ingredients	Content, %	Nutrient Level	Content
Corn	55.00	DE, MJ/kg	14.17
Wheat midding	3.50	CP, %	20.35
Phospholipid	2.00	Lys, %	1.34
Whey powder	5.00	Met+Cys, %	0.77
Extruded soybean	7.30	Thr, %	0.80
Soybean meal	18.50	Ca, %	0.95
Fish meal	5.00	TP, %	0.65
Dicalcium phosphate	1.00	AP, %	0.48
Limestone	1.10		
NaCl	0.10		
L-Lysine HCl	0.35		
DL- methionine	0.15		
Vitamin-mineral premix ^1^	1.00		
Total	100		

^1^ Supplied the following per kg of diet: Vitamin A, 10,000 IU; vitamin D3, 400 IU; vitamin E, 10 mg; pantothenic acid, 15 mg; vitamin B6, 2 mg; biotin, 0.3 mg; folic acid, 3 mg; vitamin B12, 0.009 mg; ascorbic acid, 40 mg; Fe, 150 mg; Cu, 130 mg; Mn, 60 mg; Zn, 120 mg; I, 0.3 mg; and Se, 0.25 mg.

**Table 2 animals-12-01609-t002:** Effects of *B. licheniformis* on growth performance and diarrhea rate in piglets.

Item ^1^	Diet			SEM	*p-*Value
	CON	BL500	BL1000		
Initial BW, kg	8.938	9.428	8.7050	0.327	0.714
Final BW, kg	16.482	19.045	17.521	0.613	0.251
ADG, kg	0.269 ^b^	0.343 ^a^	0.315 ^ab^	0.013	0.023
ADFI, kg	0.475	0.569	0.524	0.018	0.079
F:G	1.765	1.656	1.664	0.028	0.216
Diarrhea rate, %	9.52 ^a^	7.54 ^b^	5.56 ^b^	0.006	0.014

Note: ^a,b^ Means with different superscripts in the same row differ significantly (*p* < 0.05). ^1^ CON, control. BL, *Bacillus licheniformis*. BW, body weight. ADG, average daily gain. ADFI, average daily feed intake; F:G, feed/gain ratio. BL500, dietary supplement with 500 mg/kg *B. licheniformis*, BL1000, dietary supplement with 1000 mg/kg *B. licheniformis. n* = 6.

## Data Availability

The data presented in this study are available on request from the corresponding author.

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
