# Peer review of "Effects of Bacillus licheniformis on Growth Performance, Diarrhea Incidence, Antioxidant Capacity, Immune Function, and Fecal Microflora in Weaned Piglets"

_animals, 2022, doi:10.3390/ani12131609_

Round 1

Reviewer 1 Report

Review to: “Effects of Bacillus licheniformis on growth performance, diarrhea incidence, antioxidant capacity, immune function, and fecal microflora in weaned piglets”

Dear authors,

Thank you for the additional paper on probiotics in animal nutrition. In my opinion the manuscript is worth publication. However, the English language has to be revised by a native speaker.

Introduction:

p.2, ll.56-57: “Dietary supplemented with B. licheniformis can increase digestibility and fecal Lactobacillus counts and reduce fecal toxic gas emission in growing-finishing pigs” Please correct this sentence: E.g. “The supplementation of the diet with B.licheniformis……”

Please include a clear hypothesis into the introduction.

Materials and Methods

Sample collection:

p.3, l.83: Why was one piglet only chosen for sample collection of blood, etc? Samples from how many animals were collected in total at one collection time?

p.4, l.106: please add town and country to the Gas Chromatography System

p.4, l.107: Please correct the reference according to authors guidelines

p.4, l.115: please add town to the DNeasy Power Soil Kit

p.4, l. 119: please add the country to the Illumina Miseq platform

p.4, l. 127; 130: please add the town to SPSS software and Prism 8 software

Discussion

Please add to the discussion a chapter discussing the better effect of the lower dose of B. licheniformis. Which mechanism could cause the lower weight gain and improvement of the immune system in piglets fed the higher dose of B. licheniformis.  

Review to: “Effects of Bacillus licheniformis on growth performance, diarrhea incidence, antioxidant capacity, immune function, and fecal microflora in weaned piglets”

Dear authors,

Thank you for the additional paper on probiotics in animal nutrition. In my opinion the manuscript is worth publication. However, the English language has to be revised by a native speaker.

Introduction:

p.2, ll.56-57: “Dietary supplemented with B. licheniformis can increase digestibility and fecal Lactobacillus counts and reduce fecal toxic gas emission in growing-finishing pigs” Please correct this sentence: E.g. “The supplementation of the diet with B.licheniformis……”

Please include a clear hypothesis into the introduction.

Materials and Methods

Sample collection:

p.3, l.83: Why was one piglet only chosen for sample collection of blood, etc? Samples from how many animals were collected in total at one collection time?

p.4, l.106: please add town and country to the Gas Chromatography System

p.4, l.107: Please correct the reference according to authors guidelines

p.4, l.115: please add town to the DNeasy Power Soil Kit

p.4, l. 119: please add the country to the Illumina Miseq platform

p.4, l. 127; 130: please add the town to SPSS software and Prism 8 software

Discussion

Please add to the discussion a chapter discussing the better effect of the lower dose of B. licheniformis. Which mechanism could cause the lower weight gain and improvement of the immune system in piglets fed the higher dose of B. licheniformis.  

conclusion: 

Please take into account the different reaction of the animals to the differenrt doses of B. licheniformis

Author Response

Response to Reviewer 1 Comments

Point 1: p.2, ll.56-57: “Dietary supplemented with B. licheniformis can increase digestibility and fecal Lactobacillus counts and reduce fecal toxic gas emission in growing-finishing pigs” Please correct this sentence: E.g. “The supplementation of the diet with B.licheniformis……”

Response 1: Thank you so much for your valuable suggestion. We have corrected the sentence p.2, ll.56-57.

Point 2: Please include a clear hypothesis into the introduction.

Response 2: We have added a clear hypothesis into the introduction on lines 68-70

Point 3: p.3, l.83: Why was one piglet only chosen for sample collection of blood, etc? Samples from how many animals were collected in total at one collection time?

Response 3: We’re sorry that we didn't write the sample number accurately, in fact, we have collected 18 piglets’ samples at one time.

Point 4: p.4, l.106: please add town and country to the Gas Chromatography System

Response 4: l.128, we have added town and country to the Gas Chromatography System.

Point 5: p.4, l.115: please add town to the DNeasy Power Soil Kit

Response 5: l.137, we have added town to the DNeasy Power Soil Kit

Point 6: p.4, l. 119: please add the country to the Illumina Miseq platform

Response 6: l.141, we have added the country to the Illumina Miseq platform

Point 7: p.4, l. 127; 130: please add the town to SPSS software and Prism 8 software

Response 7: l.151, 154, we have added the town to SPSS software and Prism 8 software

Point 8: p.4, l.107: Please correct the reference according to authors guidelines

Response 8: l.129, we have corrected the reference according to authors guidelines.

Point 9: Please add to the discussion a chapter discussing the better effect of the lower dose of B. licheniformis. Which mechanism could cause the lower weight gain and improvement of the immune system in piglets fed the higher dose of B. licheniformis.

Response 9: We have discussed the better effect and possibly mechanism of lower dose of B. licheniformis in l.263-275 and l.293-297.

Point 10: Please take into account the different reaction of the animals to the differenrt doses of B. licheniformis.

Response 10: Thank you for your suggestion, we have added the discussion about the dose of B. licheniformis, and we have added the doses of B. licheniformis in the conclusion.

Reviewer 2 Report

The topic of this study is very interesting and within the scope of ANIMALS, and would also be interesting for the field of animal husbandry industry/animal nutrition sciences. The study design was suitable. The results are also very interesting. However, I noticed that the discussion part is weak. It seems that the authors have not very well combined the results in discussion, and parts/paragraphs are more like isolated and independent from each other. Growth and diarrhea are two of the most important findings in this study, the authors should discuss more on why the pigs have such problems at weaning or after weaning, and how the supplementation could affect those parameters and how these changes would benefit pigs. Not only just list the other findings and repeat the present results in the discussion part. When writing the discussion, please do not discuss the results in part to part. The authors may need to discuss the results more closely, especially when discussing the growth and diarrhea results (combine the plasma and microbiota results…). In my opinion, the discussion should be intensively revised while the writing should also be optimised. Following are the detailed comments. 

Line 20 better to have a sentence showing the research backgrounds 

Line 22 n = 180 

Line 23 500 mg/kg body weight, same for 1000 mg/kg…. 

Line 39 Early weaning is… not clear enough, what does early weaning referee? 

Line 54 I feel confusing how the repair of imbalance of intestinal flora can lead to improvement of food/feed safety… 

Line 58 it may be better to have one or two sentences mentioning the currently research progress of B. licheniformis in weaned piglets (if exist) or mention the lack of such studies. 

Line 65 if I am not misunderstanding here, this sentence meant that this study was carried out according to the regulation by the Animal Care…..University? 

Line 66 in the introduction, it was mentioned that early weaning is a stress… however, I missed the weaning age of the experimental piglets, were they weaned at 28 d of age? Please clarify here. 

Line 66 four-week-old 

Line 94 the exact LOT No. or product info are needed for each mentioned parameter. This is essential for repeat of experiment. 

Line 99 see the comment above 

Line 126, considering the n=6 for plasma and other samples, have a power analysis been done, or have the normality of data distribution and equality of variance been checked in this case? 

Table 2, I guess the unit for ADG was incorrect… 

Line 152 …that the supplemental dose of 500 mg/kg was better…, better to delete, no discussion or subjective conclusions in Result part. 

Tables and figures, n numbers are missing. 

Line 238 the growth was improved whilst feed intake and F:G were not affected by B. licheniformis, this should be discussed here: how B. licheniformis could improve growth and why the intake and F:G were not affected. Not just list the other studies… 

Line 246 I would expect the authors to discuss more about the mechanisms that how B. licheniformis could reduce diarrhea in pigs, in combination of other results in this study if needed. 

Line 256 here I would expect the authors to discuss more about the factors of weaning-leaded/associated oxidative stress and how the parameters were affect by B. licheniformis (or potential possibilities for the effects?) and how could these changes benefit pigs. Not only just list the other studies and repeat the present results. The same for the next 3 paragraphs in discussion…

Author Response

Response to Reviewer 2 Comments

Point 1: Line 20 better to have a sentence showing the research backgrounds

Response 1: Thank you so much for your valuable suggestion. We have added a sentence showing the research backgrounds in the abstract on Line 22.

Point 2: Line 22 n = 180;

Response 2: Line 25, we have corrected the word n=180.

Point 3: Line 66 four-week-old;

Response 3: Line 72, we have corrected the word four-week-old.

Point 4: Line 58 it may be better to have one or two sentences mentioning the currently research progress of B. licheniformis in weaned piglets (if exist) or mention the lack of such studies;

Response 4: Line 68, we have addded one sentence about research status.

Point 5: Line 65 if I am not misunderstanding here, this sentence meant that this study was carried out according to the regulation by the Animal Care…..University?

Response 5: Line 77, we have corrected the sentence that the current study was carried out according to the regulation by the Animal Care and Use Committee of the Zhejiang A&F University.

Point 6: Line 54 I feel confusing how the repair of imbalance of intestinal flora can lead to improvement of food/feed safety…;

Response 6: Line 58, We have corrected the questions you mentioned aboved.

Point 7: Table 2, I guess the unit for ADG was incorrect…;

Response 7: Table 2, we have corrected the unit for ADG.

Point 8: Line 152 …that the supplemental dose of 500 mg/kg was better…, better to delete, no discussion or subjective conclusions in Result part;

Response 8: Line 170, we have deleted the subjective conclusion.

Point 9: Tables and figures, n numbers are missing.

Response 9: We have added the n numbers in tables and figures.

Point 10: Line 23 500 mg/kg body weight, same for 1000 mg/kg….

Response 10: Lines 25-27, we have correted the presentation of the dose of B. licheniformis we added.

Point 11: Line 39 Early weaning is… not clear enough, what does early weaning referee?

Response 11: Early weaning refers piglets weaned at 21-28 days of age.

Point 12: Line 66 in the introduction, it was mentioned that early weaning is a stress… however, I missed the weaning age of the experimental piglets, were they weaned at 28 d of age? Please clarify here.

Response 12: Line 77, the experimental piglets were weaned at 28 days of age. Weaning is a stress, we are exploring beneficial feed additives to reduce the damage caused by stress in piglets.

Point 13: Line 94 the exact LOT No. or product info are needed for each mentioned parameter. This is essential for repeat of experiment. Line 99 see the comment above

Response 13: We have added Lot number of each mentioned parameter on Line 113-125.

Point 14: Line 126, considering the n=6 for plasma and other samples, have a power analysis been done, or have the normality of data distribution and equality of variance been checked in this case?

Response 14: We have added the methods on Lines 149-152. All data were checked by the equality of variance with normality distribution.

Point 15: Line 238 the growth was improved whilst feed intake and F:G were not affected by B. licheniformis, this should be discussed here: how B. licheniformis could improve growth and why the intake and F:G were not affected. Not just list the other studies…

Response 15: We have discussed how B. licheniformis could improve growth on Lines 256-266.

Point 16: Line 246 I would expect the authors to discuss more about the mechanisms that how B. licheniformis could reduce diarrhea in pigs, in combination of other results in this study if needed.

Response 16: We have dicussed pathogen invasion and environmental stress affect the incidence of diarrhea by changing the composition of intestinal flora, while B. licheniformis can reduce the diarrhea incidence by increasing the level of Clostridium_sensu_stricto_1 on Lines 257-259 and 338-342.

Point 17: Line 256 here I would expect the authors to discuss more about the factors of weaning-leaded/associated oxidative stress and how the parameters were affect by B. licheniformis (or potential possibilities for the effects?) and how could these changes benefit pigs. Not only just list the other studies and repeat the present results. The same for the next 3 paragraphs in discussion…

Response 17: We have added how the parameters were affect by B. licheniformis and how could these changes benefit pigs in the discussion on Line 277-283.

Reviewer 3 Report

The authors should review previous reports for an effective amount of B. licheniformis to be used in piglets. In addition, a comparison with other prebiotics commonly used in pigs is provided.

It was not clear how to feed B. licheniformis to the piglets.

Please specify the location where blood samples were taken on line 84.

Please explain how to calculate diarrhea percentage because it was confusing.

According to SEM, the means of ADG between dietary treatments in Table 2 were not statistically significant. Please double-check SEM.

Figure 2, please include a letter to indicate any significant differences in IgG and TNF.

It is preferable to cite references that involve pigs rather than broilers. Lines 232-238

Please elaborate on how B. licheniformis works to reduce diarrhea. Lines 239-246

If possible, please elaborate on how B. licheniformis creates an environment conducive to the growth of specific bacteria such as Clostridium spp.

Finally, based on the results, please specify a suitable level that improves growth performance

Author Response

Response to Reviewer 3 Comments

Point 1: The authors should review previous reports for an effective amount of B. licheniformis to be used in piglets. In addition, a comparison with other prebiotics commonly used in pigs is provided.

Response 1: Thank you so much for your valuable suggestion. We have added relevant content in introduction on Lines 51-67.

Point 2: It was not clear how to feed B. licheniformis to the piglets.

Response 2: We have described the way we fed B. licheniformis to the piglets on Lines 77-78.

Point 3: Please specify the location where blood samples were taken on line 84.

Response 3: We have added the location where blood samples were taken on Lines 91-92.

Point 4: Please explain how to calculate diarrhea percentage because it was confusing.

Response 4: We have added a formula to calculate diarrhea rate on line 109.

Point 5: According to SEM, the means of ADG between dietary treatments in Table 2 were not statistically significant. Please double-check SEM.

Response 5: We have checked the SEM in Table 2.

Point 6: Figure 2, please include a letter to indicate any significant differences in IgG and TNF.

Response 6: We found there are no significant differences in IgG and TNF, so we didn’t include a letter to them. We refered to the following literature from the magazine: Choi, J.; Tompkins, Y.H.; Teng, P.-Y.; Gogal, R.M., Jr.; Kim, W.K. Effects of Tannic Acid Supplementation on Growth Performance, Oocyst Shedding, and Gut Health of in Broilers Infected with Eimeria Maxima. Animals 2022, 12,1378. https://doi.org/10.3390/ ani12111378

Point 7: It is preferable to cite references that involve pigs rather than broilers. Lines 232-238

Response 7: We have cited references that involve pigs.

Point 8: Please elaborate on how B. licheniformis works to reduce diarrhea. Lines 239-246

Response 8: We have explained that B. licheniformis reduced the pathogens in the gut through competitive rejection, and increased the uptake of nutrients in the intestine to improve growth performance on Lines 261-263, and B. licheniformis can reduce the diarrhea incidence by increasing the level of Clostridium_sensu_stricto_1 on Lines 324-328.

Point 9: If possible, please elaborate on how B. licheniformis creates an environment conducive to the growth of specific bacteria such as Clostridiumspp.

Response 9: We have explained that B. licheniformis can improved the the levels of VFAs, including butyric acid, isobutyric acid, valeric acid, and isovaleric acid in fecal content, which can create an acidic environment in the gut to inhibit some harmful bacteria, while provide favorable conditions for the growth of beneficial bacteria such as Clostridium spp on Lines 313-327.

Point 10: Finally, based on the results, please specify a suitable level that improves growth performance.

Response 10: We have specified a suitable level that improves growth performance in the discussion and conclusion.

Round 2

Reviewer 2 Report

Thanks for your hard working on revising the manuscript. I have only a few suggestions but the writing can be improved considering the style/grammar (e.g., some words were incorrect or some sentences were too complicated/replicated).

Line 82. Was the dose per kg of BW or per kg of diet?

Line 275. It is very interesting to see if there are already some papers reporting an effect of dietary B. licheniformis supplementation on gut development (e.g., the structure of villus or crypts) or function (e.g., barrier function). If there are, maybe it can also be mentioned here.

Line 281. increased = increasing

Line 282. impaired = impairing

Line 317. had = was

Author Response

Point 1: Line 82. Was the dose per kg of BW or per kg of diet?

Response 1: It was the dose per kg of basal diet.

Point 2: Line 275. It is very interesting to see if there are already some papers reporting an effect of dietary B. licheniformis supplementation on gut development (e.g., the structure of villus or crypts) or function (e.g., barrier function). If there are, maybe it can also be mentioned here.

Response 2: We have added the relative contents about B. licheniformis supplementation on gut development on Lines 378-385.

Point 3: Line 281. increased = increasing

Response 3: We have correted the word on Line 390.

Point 4: Line 282. impaired = impairing 

Response 4: We have correted the word on Line 391

Point5 : Line 317. had = was

Response 5: We have correted the word on Line 469.

Reviewer 3 Report

The majority of the comments have been addressed. However, there are some details to consider.

Because the equation for calculating diarrhea percentage was confusing, you should present it more clearly.

Please redo the statistical analysis on Table 2, as there appears to be some doubt.

Author Response

Point 1: Because the equation for calculating diarrhea percentage was confusing, you should present it more clearly.

Response 1: We have modified the representation of diarrhea incidence.

Point 2: Please redo the statistical analysis on Table 2, as there appears to be some doubt.

Response 2: Thank you for your careful review. We have re-analyzed and checked Table 2.